# A Comparison of Total Thoracoscopic and Robotic Surgery for Lung Cancer Lymphadenectomy

**DOI:** 10.3390/cancers15133442

**Published:** 2023-06-30

**Authors:** Anna Ureña, Camilo Moreno, Ivan Macia, Francisco Rivas, Carlos Déniz, Anna Muñoz, Ines Serratosa, Marta García, Cristina Masuet-Aumatell, Ignacio Escobar, Ricard Ramos

**Affiliations:** 1Department of Thoracic Surgery, Hospital Universitari de Bellvitge, Bellvitge Biomedical Research Institute (IDIBELL), L’Hospitalet de Llobregat, 08907 Barcelona, Spain; cmorenom@bellvitgehospital.cat (C.M.); imacia@bellvitgehospital.cat (I.M.); frivas@bellvitgehospital.cat (F.R.); cjdeniz@bellvitgehospital.cat (C.D.); amunozf@bellvitgehospital.cat (A.M.); iserratosa@bellvitgehospital.cat (I.S.); mgarciami@bellvitgehospital.cat (M.G.); iescobar@bellvitgehospital.cat (I.E.); ricardramos@ub.edu (R.R.); 2Department of Thoracic Surgery, Hospital Clinic, 08036 Barcelona, Spain; 3Doctoral Programme of Medicine and Translational Research, University of Barcelona, 08036 Barcelona, Spain; 4Unit of Human Anatomy, Department of Pathology and Experimental Therapeutics, Medical School, University of Barcelona, L’Hospitalet de Llobregat, 08907 Barcelona, Spain; 5Department of Preventive Medicine, Hospital Universitari de Bellvitge, Bellvitge Biomedical Research Institute (IDIBELL), L’Hospitalet de Llobregat, 08907 Barcelona, Spain; cmasuet@bellvitgehospital.cat

**Keywords:** RATS, robotic, lymph node, lymph node dissection, lymphadenectomy, non-small-cell lung cancer

## Abstract

**Simple Summary:**

Some studies have demonstrated an association between systematic lymph node dissection (LND) and more accurate disease staging compared with dissection of randomly selected lymph nodes. The aim of this study was to retrospectively compare the quality and safety of LND performed via multiport robotic-assisted thoracic surgery (RATS) versus multiport thoracoscopy (TT). A total of 246 patients were included, 85 underwent lung resection via TT, and 161 underwent lung resection via RATS. LND was assessed based on the number of lymph node stations and the number of lymph nodes removed per station, and the postoperative complications were assessed for each technique. Some authors assert that RATS allows them to perform more extensive lymph node dissection, with a higher rate of node upstaging. This comparative study could be used to help evaluate the role of the robot in the technique.

**Abstract:**

Background: Robotic-assisted thoracic surgery (RATS) is used increasingly frequently in major lung resection for early stage non-small-cell lung cancer (NSCLC) but has not yet been fully evaluated. The aim of this study was to compare the surgical outcomes of lymph node dissection (LND) performed via RATS with those from totally thoracoscopic (TT) four-port videothoracoscopy. Methods: Clinical and pathological data were collected retrospectively from patients with clinical stage N0 NSCLC who underwent pulmonary resection in the form of lobectomy or segmental resection between June 2010 and November 2022. The assessment criteria were number of mediastinal lymph nodes and number of mediastinal stations dissected via the RATS approach compared with the four-port TT approach. Results: A total of 246 pulmonary resections with LND for clinical stages I–II NSCLC were performed: 85 via TT and 161 via RATS. The clinical characteristics of the patients were similar in both groups. The number of mediastinal nodes dissected and mediastinal stations dissected was significantly higher in the RATS group (TT: mean ± SD, 10.72 ± 3.7; RATS, 14.74 ± 6.3 [*p* < 0.001]), except in the inferior mediastinal stations. There was no difference in terms of postoperative complications. Conclusions: In patients with early stage NSCLC undergoing major lung resection, the quality of hilomediastinal LND performed using RATS was superior to that performed using TT.

## 1. Introduction

Robotic assisted thoracic surgery (RATS) is increasingly used in the surgical treatment of patients with non-small-cell lung cancer (NSCLC). The technique has been demonstrated to be safe and comparable to other minimally invasive thoracoscopic surgical techniques in terms of survival, hospital stay, and morbidity and mortality [1].

Radical lymph node dissection (LND) remains an essential component of lung cancer surgery, as it contributes to prognosis and accurate staging for decisions on future treatment. Although some randomized studies and clinical trials have not been able to demonstrate a direct association between complete LND and survival [2,3,4,5,6,7], other articles such as that by the Eastern Cooperative Oncology Group (ACOSOG ZOO30) have demonstrated an association between systematic LND and more accurate staging compared with systematic lymph node sampling.

Currently it is widely accepted that systematic LND is superior to lymph node sampling for accurate disease staging; therefore, systematic LND has been adopted by many thoracic surgeons during lung cancer surgery [8].

The benefits of minimally invasive surgery, particularly in early stage NSCLC, have been widely reported, although the benefits of RATS are the subject of discussion. Minimally invasive procedures are less aggressive for the patient, meaning less pain, shorter hospital stay, and better quality of life. However, as was the case with thoracoscopy when it was first introduced [9,10], noninferiority of RATS versus other techniques must be demonstrated for the common goal: treatment of lung cancer.

Some articles report that RATS is not superior to VATS in terms of lymph node removal or upstaging [11], but other authors assert that RATS allows dissection of more lymph nodes with a higher rate of lymph node upstaging than VATS [12,13,14,15].

There are no articles comparing LND via four-port TT versus four-port RATS (without an assist port), and few comparing the postoperative complications. We sought to compare these similar techniques—both with thoracoscopic approaches—to evaluate the role of the robot in the technique.

The aim of this study was to retrospectively assess and compare the quality and safety of LND performed via multiport RATS vs. multiport TT, evaluating the number of lymph node stations and the number of lymph nodes removed per station and the postoperative complications for each technique.

## 2. Materials and Methods

In our department, the standard of care for stages I–II NSCLC is lung resection, lobectomy, or segmental resection in patients with limited lung function, always with systematic LND.

In 2012, we began a program of minimally invasive surgery with multiport thoracoscopic surgery according to the technique described by D. Gossot [16]. Since then, RATS has become the usual technique in our department, prompting us to compare these two techniques, which are very similar in terms of number of ports and approach, for the same treatment. We appreciate that there are many minimally invasive techniques; we have tried to eliminate bias in the analysis of outcomes achieved with the robot regarding LND.

### 2.1. Study Population

We reviewed 246 patients in total: 85 underwent TT surgery between September 2010 and February 2018, and 161 underwent RATS between February 2019 and March 2022. All of them had lung resection with LND for a diagnosis of lung cancer in our institution.

The reason for introducing RATS in our management plan was that, in February 2019, a robotic program was started in our center on a public basis. The robot was used with the same indications as TT to treat pulmonary carcinoma, although the availability of the machine limited the number of cases initially, and as happens when any technique is relatively new, we tried to select patients with smaller tumors that were more likely to be fully resectable, and without a history of pleuropulmonary infections that could cause severe pleural adhesions.

The two groups selected for comparison in this study form part of the early stages of the introduction of a technique in our thoracic surgery department: RATS versus TT surgery.

All patients underwent the same preoperative workup regarding operability and resectability: thin-slice computed tomography (CT), evaluating the maximum dimension and degree of consolidation of the tumor, and positron emission tomography (PET).

Patients with lymph nodes suspicious of malignancy (on CT or PET-CT with FDG uptake in the nodes) underwent endobronchial ultrasound-guided transbronchial needle aspiration preoperatively, as did patients with tumors larger than 4 cm or central tumors, even if they did not have lymph nodes suspicious of malignancy on imaging [17]. Mediastinoscopy was not performed at the time of this study.

All cases were staged according to the 8th edition of TNM for the present study. The severity of postoperative complications was classified according to the Clavien–Dindo system [18].

### 2.2. Study Variables

The study variables included the following: age, sex, comorbidities (smoking, diabetes mellitus, ischemic heart disease, chronic kidney disease, chronic obstructive pulmonary disease, peripheral vascular disease, atrial fibrillation, hypertension, and previous cancer), type of surgery performed, clinical stage, pathological stage, tumor histology, postoperative complications in hospital, 30-day postoperative mortality, and hospital length of stay (LOS) measured from the day of surgery to discharge.

Postoperative complications included the following: atrial fibrillation, pneumonia, wound infection, hemothorax, pneumothorax, chylothorax, and persistent air leak. A diagnosis of pneumonia was based on the presence of clinical and laboratory signs of lung infection, together with radiographic findings consistent with pneumonia. Wound infection was defined as the presence of positive bacterial culture from the surgical wound with clinical signs of infection. Persistent air leak was defined as the presence of a leak on or after postoperative day 5; a digital chest drainage system (Medela–Thopaz; Medela AG, Baar, Switzerland) was used in all patients. The drainage tube was removed 24 h after air leaks were no longer detectable and the drainage volume was <100 mL. The chest drain duration was calculated from the day of insertion to the day of removal.

All the patients were admitted to the hospital on the day the surgical intervention was scheduled. The management of postoperative pain consisted of serratus and intercostal plane block with non-steroidal anti-inflammatories and rescue opioids. Antithrombotic prophylaxis, in the form of subcutaneous injection of low-molecular-weight heparin, was initiated 24 h after surgery and continued until discharge.

### 2.3. Surgical Technique and Lymph Node Dissection Procedures

All patients received standard general anesthetic with one-lung ventilation using a double-lumen endotracheal tube; they were positioned in the lateral decubitus position.

The TT technique was performed according to the previous description by D. Gossot [16]. Lymph node dissection was performed with monopolar coagulation or with the Ligasure Maryland jaw and, in general, was performed after lobectomy or segmentectomy. The dissection of stations 4R, 5, 6, 7, 8, and 9 was performed systematically according to the disease side. Stations 3 and 4L were not removed, and stations 10, 11, and 12 were removed systematically, with the exception of those lymph nodes that were removed in bloc with the specimen.

RATS was performed with the 4th generation robotic system da Vinci Xi, with only 4 ports positioned in the intercostal space (7th–8th). Pulmonary resection was performed using 4 robotic arms without an assist port and conventional carbon dioxide (CO_2_), without Airseal, connected to a separate port from the optical port. The instruments used were the tip-up grasper in the 4th arm, the cadiere grasper in the left arm, and Maryland bipolar forceps in the right arm. Nonrobotic endostaplers were used to staple veins, arteries, bronchus, and fissure, as da Vinci staplers were not available at that time. The external staplers were introduced via the most anterior port, which measured 12 mm (the other ports were 8 mm), as we had previously disconnected the robotic arm. We did not have a robotic vessel sealer (Vessel Sealer Extend or Synchroseal) for dissection or sealing. The LND was conducted, unlike in TT, at the beginning of surgery, and stations 10, 11, and 12 were also included in the specimen. All lymph node stations were also checked, starting with the most apical stations, followed by mobilizing the lung without grasping it to access the paraesophageal space, pulmonary ligament, and subcarinal space.

### 2.4. Statistical Analysis

The distribution of the variables was evaluated using the Kolmogorov–Smirnov test. For quantitative variables following a non-normal distribution, the results were expressed as medians and interquartile ranges (IQR) and compared using the Mann–Whitney U test. For quantitative variables following a normal distribution, the results were expressed as means and standard deviations and compared using the Student’s *t*-test. Between-group comparisons were performed using the Chi-squared test for qualitative variables or Fisher’s exact test when needed. A kappa index between clinical and pathological stages in robotic as well as thoracoscopic patients was also performed. All the statistical tests were two-tailed, and a *p*-value ≤ 0.05 was considered statistically significant. Data were analyzed using the Statistical Software Package SPSS 28.0 (SPSS, Inc., Chicago, IL, USA).

## 3. Results

The clinical and pathological characteristics were similar in the two groups (Table 1). There were significant differences in sex, there being more men than women (*p* = 0.75). There were also differences in respiratory comorbidities in terms of COPD (*p* = 0.014) and active smokers (*p* = 0.011) but not in other clinical or pathological variables.

There were no significant differences in the type of resection, clinical stage, or pathological stage between the two groups (Table 2).

We found no significant differences between the two groups in terms of postoperative complications (Table 3).

### 3.1. Analysis of Dissected Lymph Nodes

Figure 1 shows the total lymph node counts according to technique used.

Figure 2 shows the results of the lymph node counts for the various lymph node stations. The total number of dissected lymph nodes was significantly higher in the RATS group (*p* < 0.001; Figure 1), particularly in the superior area, stations #2, #3, #4, #5, and #6 (*p* < 0.001); the inferior mediastinal area, stations #8 and #9 (*p* < 0.001); and subcarinal station #7 (*p* < 0.001). There were also significantly more dissected lymph nodes in the RATS group from hilar stations #10, #11, and #12 (*p* = 0.026) (Figure 2).

### 3.2. Lymph Node Upstaging

Table 4 shows the incidence of lymph node upstaging. In general, there were no differences between the two groups (RATS, 32; TT, 14; *p* = 0.51). We looked at changes in clinical or pathological stage, in particular changes from cN0 to pN1, cN0 to pN2, and cN1 to pN2, and found no significant differences between the two techniques.

## 4. Discussion

Minimally invasive surgery has been demonstrated to involve less postoperative pain, shorter hospital stay, and better compliance with adjuvant chemotherapy, compared with thoracotomy [19].

Within the concept of minimally invasive thoracic surgery, we can differentiate between video-assisted surgery—in which we use an assist port plus another one, two, or three additional work ports—and totally thoracoscopic surgery (TT), in which we use four ports, as described by Gossot. Both techniques, VATS and TT, have been demonstrated as comparable to open surgery or classical posterolateral (PLT) thoracotomy in the systematic assessment of lymph nodes, and no differences have been found in the number of lymph nodes or lymph node stations [20,21,22,23].

Some surgeons have suggested that, as the thoracoscopic endoscopic instruments are straight, this may lead to reduced LND in narrow anatomical areas. The da Vinci Surgical System (DVSS; Intuitive Surgical Company, Sunnyvale, CA, USA) has some advantages, among them articulated forceps with seven degrees of freedom and a three-dimensional visual field. These innovative technologies may improve the precision and quality of LND [24].

Several large-scale retrospective studies have demonstrated that dissection of fewer lymph nodes may be associated with worse prognosis due to inaccurate staging, while a higher number of dissected lymph nodes is associated with more accurate lymph node staging and better long-term survival outcomes in surgically resected NSCLC [25,26].

Mediastinal lymph node dissection has been the subject of multiple studies, and to this day, both which technique to use and the extent to which the lymphadenectomy should be performed remain debatable. Currently, there are no clear definitions in the literature regarding the extent or type of lymph node resection that should be performed in each patient [27].

The therapeutic value of lymphadenectomy can be measured by observing local recurrence rates. Several studies have shown that complete mediastinal lymphadenectomy reduces postoperative intrathoracic recurrences and metastasis in mediastinal lymph nodes. This demonstrates that lymphadenectomy improves local disease control. It is also associated with a longer disease-free period and, therefore, a positive effect on survival [28,29,30,31].

Some authors have described performing a prepulmonary resection median sternotomy in order to perform an extensive bilateral lymphadenectomy and have compared it to radical lymphadenectomy via thoracotomy, noting longer operative times, greater blood loss, and more complications [32].

Other authors advocate for extended lymph node resection using a transcervical approach (transcervical extended mediastinal lymphadenectomy, TEMLA) as a staging method and a mediastinal approach. However, it is not clear whether TEMLA improves patient selection and survival compared to complete lymphadenectomy (LND) [33,34,35,36].

Therefore, our service has followed the guidelines of the European Society of Thoracic Surgeons (ESTS) for the staging and treatment of lung cancer [37,38].

As these results are widely accepted among thoracic surgeons, removal of as many mediastinal and hilar lymph nodes as possible is considered important in lung cancer surgery. Some studies have compared lymph node counts and rates of upstaging between RATS and other approaches (VATS and/or open thoracotomy) [11,39,40,41].

In this study, we retrospectively reviewed patients with early stage lung cancer who had undergone resection and mediastinal LND via TT or RATS and compared their perioperative outcomes and outcomes related to LND, including postoperative complications, dissected lymph node count, and lymph node upstaging. In all cases, the resection performed met the criteria for complete resection [42].

Analyzing this homogeneous group of patients, we did not find significant differences in the surgical outcomes between RATS and TT, including postoperative complications in relation to LND. We removed significantly more lymph nodes in the robotic group, in terms of total number of lymph nodes and also stations, and when we analyzed the postoperative complications, there were no significant differences. Some authors describe fewer complications with RATS, along with resection of more lymph nodes, attributing this to an easier dissection of the bronchial arteries or the thoracic duct made feasible by the better maneuverability and the 3D visual field, as well as fewer injuries to nerves such as the recurrent laryngeal nerve [43].

Upstaging is a crucial aspect in the surgical treatment of NSCLC, and some authors have described lower rates of upstaging with minimally invasive techniques [15]. However, we have not found significant differences between the two groups, in line with what other authors have also reported, even when compared to thoracotomy [39,40].

These results are probably due to the advantages of the robotic system in terms of ergonomics and vision, but further comprehensive analysis is needed to verify these conclusions.

## 5. Limitation of the Study

The main limitation is that this was a retrospective study with patients from a single center, with a limited sample that may be insufficient to draw definitive conclusions.

There were four surgeons in the TT group and the same four in the RATS group. Although it could be suggested that better outcomes were the result of certain surgeons working on one technique or another, we must point out that both techniques were used by the same surgeons, who initially used the thoracoscopic technique then later transitioned to the robotic technique. There may have been learning-curve effects, but we must bear in mind that the cases studied were the first in each program. In our department, minimally invasive surgery was introduced with TT, and now RATS is the technique used in >50% of cases.

Lastly, the lymph nodes count bias is obvious, especially when compared with other series. While the removal of lymph nodes depends exclusively on the surgeon, the count depends on the pathologist. Only whole lymph nodes were counted, fragments were discarded, and the total number lymph nodes in the specimen were not counted. We judge that intragroup bias was cancelled out as the same three pathologists analyzed the samples from both groups, although there may be significant differences in number of lymph nodes if we compare the results of this study with other external groups.

## 6. Conclusions

In summary, our findings indicate that the robotic RATS approach may be more effective than a total thoracoscopic approach in terms of the number of dissected lymph nodes, without significant differences in complications between the two techniques. Following proper clinical staging according to internationally validated guidelines, no significant upstaging was recorded with RATS. However, the advantages of this technique lead us to believe that we can optimize diagnosis and, consequently, prognosis compared to other approaches if we extend the study period.

## Figures and Tables

**Figure 1 cancers-15-03442-f001:**
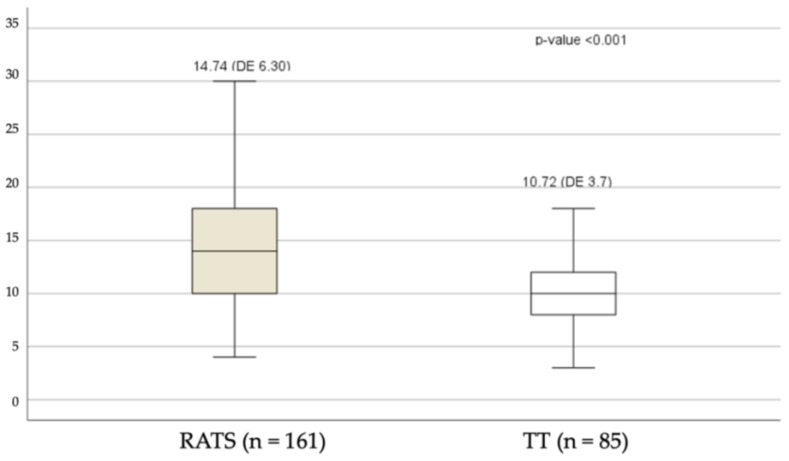
Total lymph node counts.

**Figure 2 cancers-15-03442-f002:**
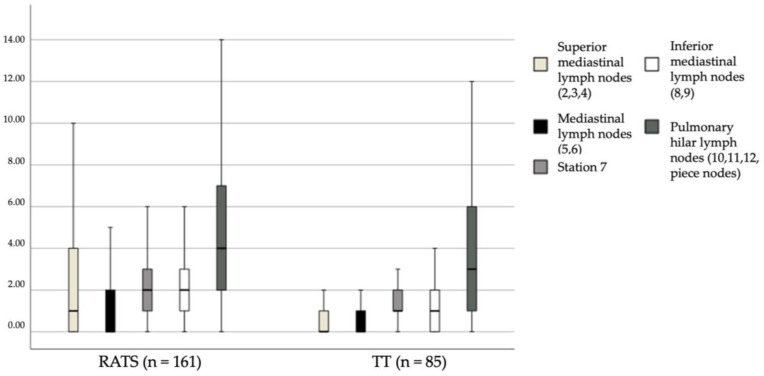
Lymph node counts for stations.

**Table 1 cancers-15-03442-t001:** Clinical and pathological characteristics of the study population.

		RATS (*n* = 161)		TT (*n* = 85)		Total (*n* = 146)		
		*n*	%	*n*	%	*n*	%	*p*-Value
Age (years)		67.2	9.40	65.47	8.04	66.66	8.98	0.066
Sex	F	62	38.5%	31	36.5%	93	37.8%	0.754
	M	99	61.5%	54	63.5%	153	62.2%	
HTN		75	46.6%	37	43.5%	112	45.5%	0.647
DM		20	12.4%	17	20.0%	37	15.0%	0.114
COPD		29	18.0%	27	31.8%	56	22.8%	0.014
CKD		9	5.6%	2	2.4%	11	4.5%	0.339
PVD		22	13.7%	8	9.4%	30	12.2%	0.332
IHD		9	5.6%	3	3.5%	12	4.9%	0.552
AF		2	1.2%	4	4.7%	6	2.4%	0.186
Non-smoker		36	22.4%	18	21.2%	54	22.0%	0.831
Ex-smoker		79	49.1%	29	34.1%	108	43.9%	0.025
Current smoker		46	28.6%	38	44.7%	84	34.1%	0.011
Previous cancer		44	27.3%	18	21.2%	62	25.2%	0.291

F: female; M: male; HTN: hypertension; DM: diabetes mellitus; COPD: chronic obstruction pulmonary disease; CKD: chronic kidney disease; PVD: peripheral vascular disease; IHD: ischemic heart disease; AF: atrial fibrillation.

**Table 2 cancers-15-03442-t002:** Type of resection, clinical stage, histology, and pathological stage.

		RATS (*n* = 161)		TT (*n* = 85)		Total		
		*n*	%	*n*	%	*n*	%	*p*-Value
Type of resection	Wedge	3	1.9%	0	0.0%	3	1.2%	0.127
	Segmentectomy	11	6.8%	2	2.4%	13	5.3%	−0.412
	Lobectomy	137	85.1%	83	97.6%	220	89.4%	
	Bilobectomy	2	1.2%	0	0.0%	2	0.8%	
	Pneumonectomy	3	1.9%	0	0.0%	3	1.2%	
	Bronchial sleeve + lobectomy	4	2.5%	0	0.0%	4	1.6%	
	Bronchial sleeve	1	0.6%	0	0.0%	1	0.4%	
Clinical stage	IA1	5	3.1%	3	3.5%	8	3.3%	0.656
	IA2	64	39.8%	39	45.9%	103	41.9%	−0.093
	IA3	46	28.6%	26	30.6%	72	29.3%	
	IB	29	18.0%	13	15.3%	42	17.1%	
	IIA	6	3.7%	2	2.4%	8	3.3%	
	IIB	11	6.8%	2	2.4%	13	5.3%	
Histology	Adenocarcinoma	104	64.6%	60	70.6%	164	66.7%	0.129
	Squamous	31	19.3%	19	22.4%	50	20.3%	
	Other	26	16.1%	6	7.1%	32	13.0%	
Pathological Stage TNM 8th	0	3	1.9%	1	1.2%	4	1.6%	0.099
	IA1	9	5.6%	7	8.2%	16	6.5%	−0.835
	IA2	47	29.2%	18	21.2%	65	26.4%	
	IA3	32	19.9%	17	20.0%	49	19.9%	
	IB	20	12.4%	23	27.1%	43	17.5%	
	IIA	9	5.6%	2	2.4%	11	4.5%	
	IIB	22	13.7%	7	8.2%	29	11.8%	
	IIIA	16	9.9%	10	11.8%	26	10.6%	
	IIIB	3	1.9%	0	0.0%	3	1.2%	

**Table 3 cancers-15-03442-t003:** Postoperative complications.

	RATS (*n* = 161)		TT (*n* = 85)		Total		*p*-Value
Complications (Yes)	*n*	%	*n*	%	*n*	%	
Air leak	57	35.4%	23	27.1%	80	32.5%	0.184
Pneumonia	8	5.0%	2	2.4%	10	4.1%	0.501
Wound infection	0	0%	0	0%	0	0%	----
Pneumothorax	0	0.0%	3	3.5%	3	3.5%	----
Atrial fibrillation	7	4.3%	1	1.2%	8	3.3%	0.269
Haemothorax	2	1.2%	0	0.0%	2	0.8%	0.546
Chylothorax	1	0.6%	1	1.2%	2	0.8%	1
Reoperation	1	0.6%	0	0.0%	1	0.4%	1
Readmission	8	5.0%	4	4.7%	12	4.9%	1
Death	2	1.2%	0	0.0%	2	0.8%	0.546

**Table 4 cancers-15-03442-t004:** Incidence of lymph node upstaging.

	RATS (*n* = 161)	TT (*n* = 85)	*p*-Value
Nodal upstages	32	14	0.515
cN0 to pN1	15	6	0.547
cN0 to pN2	17	8	0.777
cN1 to pN2	-	-	NA
Nodal downstages	-	-	NA
No changes	129	71	0.515

NA: not available.

## Data Availability

The pseudonymized data presented in this study are available on request from the corresponding author. The data are not publicly available due to legal restrictions in Spain.

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
