# Peer review of "A Comparison of Total Thoracoscopic and Robotic Surgery for Lung Cancer Lymphadenectomy"

_cancers, 2023, doi:10.3390/cancers15133442_

Round 1
Reviewer 1 Report
I appreciate the choice of the authors to share their experience about minimal invasive lymphadenectomy, also considering the small number of papers about this topic in literature.
I endorse their aim in increasing surgical techniques for higher accuracy in staging system and consequently in treatment of lung cancer.
Nevertheless, some insights and clarifications would make the analysis more exhaustive and enhance the value of the paper.
- In the abstract’s conclusions they pointed out that RATS is superior to TT in the quality of hilomediastinal LND: this sentence is only partially true because they could dissect more lymph nodes, but complications rate are similar, indeed we can see a trend of higher air leak rate in RATS group.
- The aim of the study was also evaluating the potential of RATS in incidence of lymph node up and down staging compared to TT or not? If yes, this result should be emphasized, due to its burden on prognosis and treatment.
- The period in which two techniques were compared should be overlapping, and not consequential.
- Nerve damage, especially in recurrent laryngeal nerve, should be included in postoperative complications.
- Post-operative complications should be divided basing on which station were removed: we cannot rule out in one group were dissected station associated with lower rate of complications due to their position in the thorax.
- A comparison of dissection time between two techniques might be interesting.
- In Table 4 the row cN0 to pN2 is repeated twice.
Structure and development of both grammar and syntax of the English language is appropriate with minor corrections to be made.
Author Response
A Comparison of Total Thoracoscopic and Robotic Surgery for Lung Cancer Lymphadenectomy.
The authors would like to thank the Reviewer for the interest and help revising the manuscript. We have made a number of modifications to the manuscript according to the reviewer’s suggestions.
- In the abstract’s conclusions they pointed out that RATS is superior to TT in the quality of hilomediastinal LND: this sentence is only partially true because they could dissect more lymph nodes, but complications rate are similar, indeed we can see a trend of higher air leak rate in RATS group.
I appreciate your comment and I agree with you regarding the observation of a higher air lake rate. The explanation for that concern is resolved by the learning curve. Through our experience in robotic surgery, we have observed that there is a higher air lake rate initially, which could be attributed to the change in the tactile sensation when working with robotic instruments and the difficulty in controlling force. While we agree with this observation, we did not deem it appropriate to include it in the article because we believe it diverts attention from our main objective, which is lymphadenectomy.
- The aim of the study was also evaluating the potential of RATS in incidence of lymph node up and down staging compared to TT or not? If yes, this result should be emphasized, due to its burden on prognosis and treatment.
Overstaging is not the aim of the study, but we believe that its impact on prognosis and treatment is significant enough to mention it. We would like to have a larger “n” to analyze the results, but at the moment, they do not appear to be significant.
- The period in which two techniques were compared should be overlapping, and not consequential.
I completely agree with your comment regarding the time periods, but this is due to the technological change that surgery is undergoing. In our department, we initially started with minimally invasive surgery using multiport thoracoscopy, and then we evolved to the biportal technique, although not all surgeons involved adapted to it at the same pace. Finally, the robotic approach with multiport technology emerged. The reason for not comparing them simultaneously is that currently various surgeon-dependent minimally invasive thoracoscopic techniques are being performed. We thought it would be more consistent to compare a multiport thoracoscopic approach with a multiport robotic approach, both in the learning curve stage, to reduce biases.
- Nerve damage, especially in recurrent laryngeal nerve, should be included in postoperative complications.
I share your opinion regarding the damage to the recurrent laryngeal nerve, but as this is a retrospective study, we do not have data on this complication for all patients who underwent thoracoscopic surgery. However, I can inform you that we have not recorded any cases of nerve damage in the group of patients who underwent robotic-assisted surgery.
- Post-operative complications should be divided basing on which station were removed: we cannot rule out in one group were dissected station associated with lower rate of complications due to their position in the thorax.
In all operated patients, a systematic lymph node dissection of the hemithorax undergoing resection is performed, which involves surgical exploration of all stations. Therefore, there is a risk of injury in any resection. Clearly, in left-sided resections, there is a risk of recurrent laryngeal nerve complications that should not exist in right-sided resections.
- A comparison of dissection time between two techniques might be interesting.
Thank you for your evaluation, but we did not record the exact time of lymphadenectomy in the thoracoscopic group, only in the robotic group.
- In Table 4 the row cN0 to pN2 is repeated twice.
We apologize for the mistake.

Reviewer 2 Report
It is a real professional pleasure to evaluate this paper, for which the authors deserve congratulations. My review suggestions are:
1. TT may significate "thoracotomy" for many thoracic surgeons instead of "total thoracoscopic", so the authors may choose a different abbreviation.
2. Station 2R must be presented if biopsied or not, similar to the other mediastinal stations. Also, it should be stated that 2L is not usually touched in MLD, with motivation. In figure 2 stations 2 are introduced.
3. Station 7 is not introduced into the analysis. This station subcarinal is the most important and is mandatory to be excised in lung cancer, in every patient, according to (at least) ESTS guidelines. The analysis from figures 1 and 2 must be reconsidered.
4. Discussions need to be enriched with at least several paragraphs. The importance of lymph node dissection must be presented, starting with the proposed bilateral extended mediastinal lymph node dissection through sternotomy, proposed by the Japanese, continued with VAMLA and TEMLA, and finishing with the present guidelines and practice for mediastinal lymph node dissection.
5. Complications after lymph node dissection must be separated from the other complications, in order to offer a clear description of the situation. And how these complications must be addressed.
6. Row 246 – is the authors’ impression or the actual results?
7. Conclusions may be enriched after extending the Discussions.
8. Rows 113-115 do seem not erased from the template.
Congratulations for your work!
Author Response
A Comparison of Total Thoracoscopic and Robotic Surgery for Lung Cancer Lymphadenectomy.
The authors would like to thank the Reviewer for the interest and help revising the manuscript. We have made a number of modifications to the manuscript according to the reviewer’s suggestions.
- TT may significate "thoracotomy" for many thoracic surgeons instead of "total thoracoscopic", so the authors may choose a different abbreviation.
We appreciate your feedback and understand that it may lead to confusion. However, we have decided to maintain the term "total thoracoscopic" (TT) as it is the designation given by Domenique Gossot, who was the first to develop this technique, considering the specificities of working with four ports, without an assisting port, and using specialized instruments, distinguishing it from VATS. To avoid confusion, we differentiate a thoracotomy as PLT (posterolateral thoracotomy).
- Station 2R must be presented if biopsied or not, similar to the other mediastinal stations. Also, it should be stated that 2L is not usually touched in MLD, with motivation. In figure 2 stations 2 are introduced.
Of course, we also agree with the assessment regarding lymphadenectomy in the upper mediastinal territories.
We were considering how to express the number of excised lymph nodes in a single graph, and we encountered the difficulty of showing laterality to simplify the drawing. Therefore, we decided to separate the territories into upper and lower regions, identifying station 7 because it can be reached from both hemithoraxes, and separating 5 and 6 because they can only be left-sided. In Figure 2, the stations are compared in the following way:
- Upper mediastinal lymph nodes (stations 2, 3, 4).
- Mediastinal lymph nodes (stations 5, 6).
- Station 7.
- Lower mediastinal lymph nodes (stations 8, 9).
- Hilar and pulmonary lymph nodes (stations 10, 11, 12).
You are correct that we should have separated the stations considering the 4L, and we will take that into account for future studies.
- Station 7 is not introduced into the analysis. This station subcarinal is the most important and is mandatory to be excised in lung cancer, in every patient, according to (at least) ESTS guidelines. The analysis from figures 1 and 2 must be reconsidered.
We greatly appreciate any feedback that can improve our study, and we certainly agree on the mandatory removal of station 7 and its importance according to international guidelines. In all patients, we examined all lymph node stations in the operated hemithorax, including the subcarinal station. The study records the average number of lymph nodes removed in each station, including station 7. We apologize for the oversight of not reflecting this analysis in the text, and we have modified it by adding this data (subcarinal station #7 (p < 0,001).
- Discussions need to be enriched with at least several paragraphs. The importance of lymph node dissection must be presented, starting with the proposed bilateral extended mediastinal lymph node dissection through sternotomy, proposed by the Japanese, continued with VAMLA and TEMLA, and finishing with the present guidelines and practice for mediastinal lymph node dissection.
Following your instructions, we have expanded the discussion.
“Mediastinal lymph node dissection has been the subject of multiple studies, and to this day, it remains debatable which technique or the extent of lymphadenectomy should be performed. Currently, there are no clear definitions in the literature regarding the extent or type of lymph node resection that should be performed in each patient.
The therapeutic value of lymphadenectomy can be measured by observing local recurrence rates. Several studies have shown that complete mediastinal lymphadenectomy reduces postoperative intrathoracic recurrences and metastasis in mediastinal lymph nodes. This demonstrates that lymphadenectomy improves local disease control. It is also associated with a longer disease-free period and, therefore, a positive effect on survival.
Some authors have described performing a pre-pulmonary resection median sternotomy in order to perform an extensive bilateral lymphadenectomy and have compared it to radical lymphadenectomy via thoracotomy, noting longer operative times, greater blood loss, and more complications.
Other authors advocate for extended lymph node resection using transcervical approach (transcervical extended mediastinal lymphadenectomy, TEMLA) as a staging method and mediastinal approach. However, it is not clear whether TEMLA improves patient selection and survival compared to complete lymphadenectomy (LND).
Therefore, our service has followed the guidelines of the European Society of Thoracic Surgeons (ESTS) for the staging and treatment of lung cancer.”
- Complications after lymph node dissection must be separated from the other complications, in order to offer a clear description of the situation. And how these complications must be addressed.
I appreciate the comment. Regarding complications, it is challenging to separate those directly attributed to lymphadenectomy from others. We believe that lymphadenectomy is an integral part of surgery, and that is why we have included them in a combined table. It is true that classically described complications are directly related, such as recurrent laryngeal nerve injury during level 5 dissection or phrenic nerve injuries during the resection of nodes in levels 6 or even 3. However, in this study, we aimed to demonstrate that robotic surgery is not inferior to thoracoscopic surgery in terms of quality. We were pleasantly surprised by the results, and I agree with the reviewer that it would be worthwhile to emphasize more details as suggested. However, we believe that such emphasis would be more suitable for another study to avoid diverting attention from our main objective.
- Row 246 – is the authors’ impression or the actual results?
We are referring to our impression, as lung cancer surgeons, that we would like the upstaging rates to be 0. In other words, we would like the clinical TNM to be the same as the pathological TNM because it would mean that preoperative staging is excellent.
- Conclusions may be enriched after extending the Discussions.
Following your instructions, we have modified the conclusions.
Our findings indicate that the robotic RATS approach may be more effective than total thoracoscopic in terms of the number of dissected lymph nodes, without significant differences in complications between the two techniques. Following proper clinical staging according to internationally validated guidelines, no significant upstaging was recorded with RATS. However, the advantages of this technique lead us to believe that we can optimize diagnosis and, consequently, prognosis compared to other approaches if we extend the study period.
- Rows 113-115 do seem not erased from the template.
We apologize for the error. We have made the necessary correction.

Round 2
Reviewer 2 Report
- Row 246 – is the authors’ impression or the actual results?
We are referring to our impression, as lung cancer surgeons, that we would like the upstaging rates to be 0. In other words, we would like the clinical TNM to be the same as the pathological TNM because it would mean that preoperative staging is excellent.
Reevaluation: ...in this case please explain this into the text, instead of using "our impression" in order to offer scientific language and not smth like personal opinion.
Congratulations for your work!
Author Response
Thank you once again for your assistance in reviewing our article. Following your suggestion, we have made the following modifications to the text:
"Upstaging is a crucial aspect in the surgical treatment of NSCLC, and some authors have described lower rates of upstaging with minimally invasive techniques [45]. However, we have not found significant differences between the two groups, in line with what other authors have also reported, even when compared to thoracotomy [46,47]."